# Development and evaluation of 'Sleep, Baby & You'—An approach to supporting parental well-being and responsive infant caregiving

Helen L. Ball[1]*, Catherine E. Taylor[1], Victoria Thomas[2‡], Pamela S. Douglas[3‡], the SBY working group[¶]

**1** Infancy & Sleep Centre, Department of Anthropology, Durham University, United Kingdom, **2** Dept Paediatrics, Great North Children's Hospital, Newcastle upon Tyne, United Kingdom, **3** Possums Education & Research Centre, Greenslopes, Brisbane, Australia

☯ These authors contributed equally to this work.
‡ These authors also contributed equally to this work.
¶ Membership of the SBY Working Group is listed in the Acknowledgments.
* h.l.ball@durham.ac.uk

**Data Availability Statement:** All relevant data are available at the Durham Research Online Datasets

## Abstract

Disrupted parental sleep, presenting as post-partum fatigue and perceived as problematic infant sleep, is related to increased symptoms of depression and anxiety among new mothers and fathers. Previous research indicates that UK parents would value an approach that facilitates meeting their infants' needs while supporting their own sleep-related well-being throughout their infant's first year. Six initial stakeholder meetings were held with 15 practitioners and 6 parents with an interest in supporting parent-infant sleep needs, to explore existing service provision and identify gaps. The Possums Sleep Program developed and delivered in Brisbane, Australia in a GP clinic setting, was chosen as an appropriate approach. Working collaboratively with a stakeholder group, we translated the Possums Sleep Program into an intervention that could be universally delivered in the UK via NHS antenatal and postnatal practitioners. Parent and practitioner views of the initial materials were obtained via feedback questionnaires and the tool was revised. The intervention was then field-tested by 164 practitioners who delivered it to at least 535 new parents and babies over 5 UK locations, to capture anonymous parent and practitioner views of the intervention concept, the materials, and their experiences with both. The intervention helps parents recalibrate their expectations of infant sleep development, encourages responsive parenting and experimentation to meet their infant's needs, offers parents strategies for supporting the development of their babies' biological sleep regulators and promote their own well-being, and teaches parents to manage negative thinking and anxiety that can impede sleep using the principles of Acceptance and Commitment Therapy. The 'Sleep, Baby & You' discussion tool, a 14 page illustrated booklet for parents, was field-tested and evaluated by practitioners and parents who offered enthusiastic feedback. Practitioners reported the 'Sleep, Baby & You' materials were easy for them to explain and for parents to understand, and were a good fit with the responsive parenting approaches they employed in other areas of their work. Parents who received the intervention postnatally understood the material and found the suggestions easy to follow. All parents who provided feedback had implemented one or

Archive (DRO-DATA). The following DOI has been assigned: doi: 10.15128/r20c483j404.

**Funding:** Funding for this study was received by HB (with VT & PD as named collaborators) from the Durham University ESRC Impact Acceleration Account Fund. This funding was used, in part, to employ CT as a postdoctoral research associate to work on this project, and for PD to visit Durham to work with us in 2018. HB received a Research Seedcorn Grant from Durham University to visit Possums Clinic in Brisbane, Australia in January 2017.

**Competing interests:** HB is honorary Chair of the Scientific Committee for the Lullaby Trust, the UK's leading SIDS-prevention charity. She also serves in an honorary capacity on the Board of Examiners for the Unicef UK Baby Friendly Individual Awards, and on the Board of Directors for ISPID (the International Society for the Study and Prevention of Infant Deaths). PD is Medical Director of Possums Education, a health promotion charity which sells education programs that upskill health professionals in delivery of the Possums Sleep Program. All revenue raised by Possums Education is invested in the development of research and educational materials. We confirm the above disclosures do not alter our adherence to PLOS ONE policies on sharing data and materials.

more of the suggested changes, with the majority of changes (70%) being sustained for at least two weeks. Practitioners recommended development of digital and antenatal versions and offered feedback on circumstances that might challenge effective uptake of the intervention. 'Sleep, Baby & You' is a promising tool for promoting parental attitude and behaviour-change, that aims to adjust parental expectations and reduce negative thinking around infant sleep, promote responsive infant care in the face of infant-related sleep disruption and fatigue, and support parental well-being during the first year of parenthood. Initial field-testing provided insights useful for further development and subsequent testing via a randomised trial. Support exists for incorporating 'Sleep, Baby & You' into an anticipatory, universal intervention to support parents who may experience post-partum fatigue and infant sleep disruption.

## Introduction

Sleep is a major preoccupation for new parents, with disrupted and reduced sleep, tiredness and fatigue being common during their child's infancy, requiring behavioural and mental adjustment [1]. Although post-partum fatigue (PPF) is sometimes narrowly defined as occurring within the first 6-weeks after giving birth [2], a recent review from the UK National Perinatal Epidemiology Unit [3] characterised PPF as a consequence of disrupted sleep due to night-waking infants, difficulties settling the baby and night-time feeding that up to two-thirds of women (64–67%) experience for up to 2 years post-partum. By analysing data from the 2014 UK National Maternity Survey the authors determined which UK women were most affected. Prevalence of PPF varied by maternal age, level of socioeconomic deprivation, education and parity; women reporting depression, anxiety, sleep problems and those breastfeeding were at significantly increased risk [3].

These quantitative findings reinforce the comments of women from previous focus groups exploring sleep-related experiences during the first-year post-partum [4]. Women spoke about the ways in which sleep disruption and fatigue interfered with their mental health, their emotional stability, their relationships with others, and their experiences of emotions and reactions that were out of proportion with the events around them, affecting their ability and desire to function. Anxiety and worry accompanied the inability to keep up with 'responsibilities' within and outside the home [1]. There is also ample evidence that parental fatigue and sleep disruption can lead to the development of mood disorders in both mothers and fathers [3,5–8] and exacerbate existing mental and physical health problems [9,10]. Furthermore, unaddressed parental sleep disruption can lead to negative outcomes for infants such as being medicated [11], medicalised [12], punished and abused [13]. Finding effective ways to support parents during this period is vital.

To date, approaches for helping parents who are experiencing PPF, and particularly infant-related sleep disruption (IRSD), have involved a range of interventions that have been subjected to (more or less rigorous) evaluation via randomised trials [e.g. 7, 14–22]. Commonly, interventions designed to reduce parental (usually maternal) PPF have been universal (delivered to all childbearing women) and anticipatory (delivered before PPF symptoms are diagnosed), attempting to reduce post-partum depression and self-reported fatigue [15,16]. Likewise some interventions for IRSD have been anticipatory and universal with a general focus on establishing bed-time routines, self-settling, and sleep schedules [7,17], with more elaborate approaches for older infants and toddlers combining the latter with guidance on feeding, bathing, and bed-time stories [18–21]. In contrast the traditional intervention strategy for ISD has been targeted and treatment-oriented, aiming to improve disrupted parental sleep

by eliminating infant 'sleep problems' [22–24]. Treatment-oriented interventions often involve extinction techniques applied to infants who are causing parental sleep disruption [22,23,25–27]. Indicative examples of these approaches are summarised in Table 1.

Although PPF and IRSD are common among new parents there are few sleep-related interventions that aim to support parental resilience and foreground parental well-being (rather than infant sleep) as their primary outcome. Very few aim to responsively meet infant needs across the first year and so new initiatives are needed. We report here on how we developed and field-tested new materials in collaboration with a range of interested stakeholders for implementation in a UK intervention. The materials will be used as the basis for an intervention intended to reduce the proportion of parents experiencing postpartum depression, PPF and IRSD by providing guidance and support to parents on meeting the needs of their infants, understanding their infant's sleep, managing their fatigue and sleep disruption, and maintaining their own well-being.

## Methods

### Step 1. Intervention search, selection, and familiarisation

Prior to devising an intervention from scratch, we consulted the literature to determine whether an existing programme might meet our needs. Systematic reviews and studies published prior to August 2016 were examined via a non-systematic scoping review to identify programmes employing responsivity to infant needs and promoting parent and infant well-being. Selection criteria also included a strong theoretical rationale, an evidence-informed (based on a thorough understanding of relevant research) or evidence-based (effectiveness demonstrated via randomised trial) foundation, and active usage in an English-speaking setting. Via a process of elimination that involved contacting authors of shortlisted studies, and institutions where programmes were potentially in operation, the list of interventions was narrowed down and one was chosen for in-person investigation.

**Table 1. Examples of intervention types for PPF/IRSD.**

| | |
|---|---|
| Anticipatory universal intervention for PPF | 'Wide Awake Parenting' is designed to reduce post-partum fatigue, consisting of professionally-led telephone support with a workbook, home visit, and three telephone calls [28,29]. In a randomised trial the primary outcomes captured 6-weeks post-intervention included symptoms of fatigue, depression, anxiety and stress. Women receiving the full intervention reported fewer symptoms of fatigue, depression, anxiety, and stress than mothers in the control (wait-list) condition. |
| Anticipatory universal intervention for IRSD | 'Baby Business' is designed to reduce parentally-reported infant sleep problems involving educational guidance on normal infant sleep and cry patterns, and how to develop feeding and settling routines, with a DVD, booklet, telephone consultation, and a parent group session [7,30]. In a randomised trial the primary outcome was to reduce infant sleep problems from 30% to 20% of families when infants were 4-months of age. No differences were found between intervention and control groups in caregiver report of infant sleep, crying or feeding problems or caregiver reports of depression symptoms at 4 months. |
| Targeted treatment intervention for IRSD | A randomised trial comparing a cognitive-behavioural group intervention involving gradual extinction was compared to a control group receiving infant sleep safety information. Intervention parents received sleep education and information on 'controlled comforting' between 6–8 months at dedicated sessions with a follow-up call from a public health nurse [26]. The primary outcomes were to reduce the number of nightly wakes recorded by actigraphy and sleep diaries, and to reduce parental reports of sleep problem severity. No significant differences were found in night-waking between the intervention and control groups. The intervention group had significantly fewer parentally assessed severe sleep problems. |

The Possums Sleep Program (PSP) was the only intervention meeting all of the selection criteria in October 2016. The PSP was developed by Dr Pamela Douglas and has been delivered to parents in the Possums Clinic, Brisbane, from 2011. Dr Koa Whittingham manualised the program and integrated it with Acceptance and Commitment Therapy in 2014 [31,32]. The Possums Sleep Program is one part of a suite of programs for parents and babies known as Neuroprotective Developmental Care (NDC), or the Possums programs. It continues to be delivered to parents by NDC accredited practitioners at the Possums Clinic in Brisbane and in multiple other clinics in Australia.

The theoretical basis for the PSP is documented in detail elsewhere [31–34], and so we summarise it briefly here. Whittingham & Douglas [31] published the theoretical frame for a paradigm shift in parent-baby sleep management, based on an evolutionary understanding of infant needs, and integrating interdisciplinary sleep science and contextual behavioural science (a 'third wave' behaviourism]. The PSP translates this theoretical foundation into clinical practice. The program is intended to educate parents in basic sleep science and how this relates to their infant's sleep development. It offers strategies for optimising healthy function of the biological sleep regulators, to protect against excessive night-waking. It supports values-clarification and empowers parents to adjust their expectations and to experiment with how they might meet their babies' needs. It supports cued care according to the most recent neuroscience and psychological attachment research, with the aim of promoting secure attachment, adequate sleep, and parent-infant enjoyment [34].

With permission from the Possums Medical Director, HB spent a 6-week period in early 2017 in Brisbane where she observed (with patient permission) GP consultations, follow-up group sessions, parent drop-in sessions, and practitioner training workshops where the PSP was delivered. She also reviewed the PSP parents' workbook (64 pages) and DVD (75 minutes) together with all previous publications relating to the programme's development and implementation. During her visit the first author conducted focus groups and a qualitative online survey with parents who had used the PSP, resulting in a publication exploring parents' experiences with the PSP co-authored with the Possums team [34]. The parent evaluation and Ball's observations of the PSP in practice confirmed that the approach taken by this programme was consistent with the responsive approach to the parenting of infants recommended by Unicef UK Baby Friendly Initiative which accredits maternity, neonatal and community health services for parents and babies in the UK, and would be appropriate for development into a UK-based intervention. However, the delivery format, parent-facing materials, and training materials required adapting and translating for a UK NHS setting.

## Step 2. Stakeholder consultation

During Spring 2017 HB (academic lead) and VT (clinical lead) invited interested stakeholders in the greater Newcastle-upon-Tyne area to share their views on the needs of parents regarding infant sleep, parental fatigue and sleep disruption, the nature of existing service provision and potential gaps, the tools and training available to practitioners, and the value of translating the PSP for a new UK intervention. Six stakeholder discussion groups, each involving 3 to 15 participants (26 in total, see Table 2), were attended by paediatricians, perinatal psychologists, health visitors, infant feeding co-ordinators, community support workers, third sector parent support volunteers and staff, private practice parenting educators, and academic staff. Participants discussed the availability of existing tools within the NHS, third sector, and private practice, for supporting new parents around sleep-related issues. Stakeholders confirmed there was a gap in UK service provision addressing parental fatigue and sleep disruption during infancy. Parental guidance offered by practitioners came from personal experience, and a brief guide

**Table 2. Attendee numbers for stakeholder meetings.**

| Meeting | Number of participants |
|---|---|
| 1 | 3 (2 providers 1 parent) |
| 2 | 8 (6 providers 2 parents) |
| 3 | 3 (2 providers, 1 parent) |
| 4 | 5 (3 providers 2 parents) |
| 5 | 15 (10 providers, 5 parents) |
| 6 | 15 (13 providers, 2 parents) |

regarding extinction techniques offered by the Solihull Approach (a workforce programme for care professionals working with families [35]). As most community services in the region encouraged responsive parenting, extinction methods were considered inappropriate. The outcomes from the stakeholder consultation led to funding from Durham University to develop intervention materials and conduct field-testing.

## Step 3. Development of the 'Sleep, Baby & You' intervention materials

A working group comprising the research team and 10 stakeholders was convened to develop the intervention. The PSP materials were reviewed by the research team and nine key messages were extracted (Table 3). One of the core PSP principles, to identify and address unmanaged infant feeding problems, is beyond the scope of this intervention so was intentionally not included. Guidance for practitioners implementing the intervention will include being alert to infant feeding problems and signposting to appropriate support.

HB & CT translated the PSP into a postnatal discussion tool (booklet) named 'Sleep, Baby & You' (SBY) based on a distillation of the information contained in the Possums Sleep Workbook, the Possums DVD, and information captured by HB during observations at Possums Clinic, with consultation from PD. A graphic artist was commissioned to illustrate the key points of the intervention, and an initial draft of the discussion tool was produced. Stakeholders shared drafts with colleagues and collated their feedback from completed evaluation forms, while HB and CT shared draft materials and collated feedback from a wider group of parents and practitioners online using the JISC Online Surveys (a secure higher education online survey platform). Via an iterative process we refined the text of the discussion tool and agreed amendments to the visual materials.

Detailed feedback on the SBY postnatal discussion tool from 64 respondents (40 practitioners and 24 parents) was discussed at four stakeholder working group meetings. Of those who

**Table 3. Possums Sleep approach key principles.**

| | |
|---|---|
| 1. | Provide cued care–respond appropriately to infant cues (responsive parenting) |
| | [Identify and address unmanaged infant feeding problems] |
| 2. | Find ways to dial baby down early and keep self-dialled down–experiment and become expert |
| 3. | Promote healthy functioning of biological sleep regulators (sleep pressure and circadian cycle) |
| 4. | Identify parenting values and encourage enjoyment of baby (Acceptance & Commitment Therapy) |
| 5. | Allow baby to observe and engage with life outside the home |
| 6. | Avoid obsession with day-time naps–naps happen on the go |
| 7. | Accept that sleepiness after eating is a biological cue–let baby sleep |
| 8. | Be flexible—fit baby into everyday life (mindfully) |
| 9. | Accept that the mind often generates unhelpful thoughts in situations of stress or fatigue, which don't need to be believed, and return attention to present moment |

reviewed the materials 78% felt they gained an excellent or good understanding of the SBY approach, while 14% felt their understanding was OK, and 3% poor. Opinions regarding the illustrations were also positive with 66% indicating excellent or good, 17% OK, and 6% poor. Suggestions for amending the materials indicated we should:

a. Ensure images reflect the composition and experiences of a wide cross-section of the target population including ethnicity, age, sexual orientation of individuals depicted. Respondents requested images include tattoos, piercings, hijabs, non-natural hair colours, and settings to reflect urban streets, playgrounds and community centres rather than leafy lanes, parks, and baby massage classes.

b. Amend the text to further simplify language, include shorter sentences, reduce the use of polysyllabic words and pitch to an overall year 5 (age 10) reading age. Introductory information to be added on normal infant sleep and what to expect.

c. Ensure images avoid promoting the use of baby bottles and dummies/ soothers.

d. Increase text size and highlight key messages via use of call-out boxes.

## Step 4. Field-testing and evaluation

Approval to field-test and evaluate the SBY materials was granted by Durham University Research Ethics Board (26/09/18), and permission was received from service managers in two northeast NHS Trusts to field-test the intervention within their service and training sessions for 23 practitioners were held at two locations in northeast England in October 2018 (evaluation phase 1). We subsequently broadened field-testing to two further NHS Trusts, and one peer support organisation who volunteered to assist with evaluation and and 8 training sessions for 164 practitioners were held at these three locations (evaluation phase 2).

In both phases of field-testing practitioners were trained by HB to deliver the intervention during half-day sessions. At each session staff were provided with the background rationale for the intervention, introduced to the principles of the PSP, and feedback from parents in Australia. The SBY discussion tool was introduced and each component was discussed in detail with the underlying evidence and rationale. Practitioners were encouraged to ask questions and to share issues parents raised about infant sleep during consultations. Attendees viewed a film demonstrating the approach in a one-to-one setting and discussed two case-study examples. Details of the evaluation questions were shared and used to help practitioners consider how they might incorporate SBY in their practice. All attendees received the SBY discussion tool, a practitioner's guide explaining the rationale behind each aspect of the approach, and a 'key steps' summary. The training session presentations, and links to a website containing online resources were circulated to staff electronically. Several weeks after the training session the research team contacted the practitioners for feedback. In phase 1 staff feedback was captured via telephone interviews and practitioners also asked parents they worked with to provide anonymous feedback of their experience of SBY via a JISC online or telephone survey. Staff feedback for phase 2 was captured anonymously via a separate JISC online survey due to practitioner workload constraints that precluded telephone interviews.

## Results

### Phase 1 evaluation outcomes

The objectives of the evaluation were to explore whether SBY helps health practitioners to support families with infant sleep issues, and to find out how families respond to this approach.

Training sessions were held in October 2018, and telephone interviews were conducted in the following January with 74% (17/23) of the health visitors who received training. Of the remaining six, two had changed posts, three were absent from work, and one was uncontactable. Consent to contact was received from 16 families who received the SBY intervention materials from their health practitioner. Anonymous feedback was collected from 12 families (one parent per family) via phone or online survey in December 2018 and January 2019, 50% providing feedback 1–2 weeks after receiving SBY, and 50% after five or more weeks had elapsed. As the feedback was provided anonymously parents were not asked to provide any identifying information such as their gender, or their relationship to the child (e.g. mother/father). All interviews were audio recorded with consent, and transcribed. NVivo 11 software was used to code and sort the responses.

**Practitioner feedback.** All of the practitioners interviewed responded positively to the SBY materials and were enthusiastic about the concepts and suggestions contained within the SBY programme finding it to be empowering to parents, alleviating their stress and worries about infant sleep, and providing reassurance. Practitioners felt SBY also empowered and equipped them to advise parents about infant sleep concerns. Nearly all (15/17) practitioners responding in phase 1 appreciated having a non-sleep training option to offer parents. SBY was considered a 'common sense approach' that fit well with other initiatives to promote health and wellbeing within families. Comments made throughout the interviews demonstrated that practitioners appreciated the flexible non-prescriptive approach, the focus on normalising infant sleep, and confirmed SBY filled a gap in current service provision.

"I'm using it all the time, I think it's amazing, there's nothing that you teach that could do any harm so why wouldn't you implement it? It's just so common sense, this is definitely the future" [HV02]

"The only thing we've got is Solihull which is about the disappearing chair or controlled crying and we get a lot of people who don't want to do the controlled crying because they can't leave their babies to cry. . .so yes, it absolutely fills a gap [in service provision] . . ." [HV10]

Eight of the phase 1 practitioners had delivered SBY as recommended with some of their clients, talking through each page of the discussion tool in a single sitting before giving it to them to keep. In doing so they considered the SBY leaflet to be a useful tool to help explain concepts to parents (i.e. sleep pressure) however HVs did not report how long they spent when delivering the SBY information in full, which would have been useful for us to capture. Lack of clients specifically seeking help with infant sleep was the main reason for not following the preferred delivery format. All practitioners reported incorporating SBY ideas and principles into their informal discussions with parents (both with and without perceived infant sleep problems). Potential barriers to the recommended delivery format that emerged were when practitioners felt it was more appropriate to 'drip-feed' the information than deliver it in a single setting, and that some families were seeking a 'magic wand' solution for sleep/fatigue problems and therefore were unreceptive to any approach that encouraged parental behavioural and/or cognitive changes. For other families the use of any leaflets was considered unproductive, particularly by those practitioners working in areas with many teenage parents and socioeconomic deprivation. Providing SBY via videos was considered desirable for these clients.

"Some people want the magic wand and want their babies to sleep through from 3 months, so we can say well actually babies don't always sleep from 3 months . . . they feed the

hungrier baby food to try to get them to sleep because they think that babies should be sleeping from 3 months . . . they get frustrated, that's the clientele we are working with, if their babies are not upstairs and asleep allowing them to have their own time then they get frustrated with their babies" [HV04]

"With lots of mums I work with it would be too much for them, too complicated, but the general ideas, you know building up sleep pressure, things like that, is good, it's just getting those ideas across to those mums in different ways" [HV05]

"We don't give out as many leaflets now so apps work well, also women can keep going back to the app and they never lose their phones whereas they might lose the leaflet." [HV16]

Overall, however, practitioners who offered feedback in phase 1 reported that parents were receptive to, and understood, the SBY information. Parents experiencing infant sleep problems, first-time parents, and those with younger infants were felt to be most receptive. All practitioners interviewed were keen to see SBY being used and developed in the future, offering ideas for extending to older children, introducing antenatally, regular reinforcement to challenge popular myths, and creating digital resources to complement the discussion leaflet.

"The way that people think about sleep and the way they talk about sleep, it's just turning the whole thing around, it's difficult to get people to see new concepts, they say oh my mam said this and my gran says this, you know, sometimes it's difficult" [HV06]

"Planting the seed antenatally is important, letting people know that they shouldn't expect that baby should start sleeping through at 3 months but also giving them the tools to be able to deal with it, starting the day, getting out, fresh air, a lot of parents seemed to take that idea on board when we talk about it." [HV07]

**Parent feedback.** Twelve parents providing anonymous feedback had babies aged 2–12 months (mode = 6–8 months) and reported frequent night waking, difficulty settling the baby, and feeling stressed or worried about their baby's sleep as the most common reasons for seeking help from their health practitioner. Eight of these parents reported that their practitioners had talked them through the SBY materials while four had received the SBY leaflet only (not the recommended implementation approach). Five participants were first-time parents.

All components of the SBY approach were rated helpful or somewhat helpful by parents in improving their understanding of infant sleep, and somewhat or very helpful in thinking about what they could do about the problem they wanted to address. In comments parents specifically appreciated obtaining a better understanding of normal infant sleep patterns and the build-up of sleep pressure, how acting calm helps to keep babies calm, and the importance of consistency. Parents also found talking about negative thinking and mindfulness helped to improve their own sleep.

"Knowing that this [night-waking] is normal is helpful." [ID 42643105]

"The nap info made sense and I'm still trying it, I didn't realise how little things could affect him." "[I have found] if I'm not stressed he seems to also be calmer." [ID 42683190]

"[My] health visitor in the south never gave me any info, [I] moved to Newcastle and the health visitor here told me all this new info! I understand a lot more about his sleep, how to help him a bit more and to be consistent with it." [ID 42683190]

"It is written in a non-judgemental way that is very accessible. Crucially it contained lots of information about what constitutes 'normal' infant behaviour that would be invaluable to new mums." [ID 42643105]

All parents who provided feedback implemented one or more behavioural changes suggested by SBY with 70% sustaining them for more than 2 weeks. Improvements in parents' frequent night waking and feeling stressed about their baby's sleep were the most frequently reported positive outcomes (10/12). Parents confirmed SBY had made night-times easier (10/12), and daytimes more enjoyable (11/12) with their baby.

"Lovely nursery nurse Harriet really helped me, one chat to her with the leaflet took a huge weight off my shoulders, I've stopped trying to fix things that aren't broken, I just needed someone to tell me that my babies routine is normal and to stop worrying all the time. Best leaflet I've ever read. Thank you." [ID 41787468]

"[I] had been experiencing sleep problems mainly related to [child's] illnesses. The most useful part was the information about mindfulness. Overall, [I] really liked the leaflet and found both the leaflet and the discussion with health practitioner very useful." [ID 41787674]

"The leaflet with all information made my day, I had been doing everything in the leaflet until I believed we needed our own space and baby should be in his own room/bed. However this made everything seem positive again, we are still co-sleeping and baby naps around our day and we are a happy relaxed family again." [ID 42666435]

## Phase 2 evaluation outcomes

The aim of the phase 2 was to evaluate SBY with more widespread parent and practitioner samples. As noted above six training sessions for 164 practitioners were held at three locations in northeast England and East Midlands between March and July 2019. Practitioners were invited via email to provide feedback via an anonymous JISC online survey. Invitations were emailed on average 9.3 weeks after the training with 2–4 weeks allowed for completion.

**Practitioner evaluation of approach.** Responses were received from 93/164 practitioners (57% response rate) of which 85% (79/93) of respondents worked in community settings, the remainder being clinical staff (3%; 3/93), community support workers (2%; 2/93), peer supporters (6.5%; 6/93), public health staff (1%; 1/93) and programme managers (2%; 2/93). Respondents again appreciated SBYs 'common sense' approach and felt it fitted well with the current approaches they used in their work (e.g. breastfeeding support, motivational interviewing, and the Solihull approach). Free text comments indicated that practitioners appreciated that SBY did not employ sleep training methods, and liked SBYs flexible and adaptable style. Practitioners felt that the terminology and visual explanations within the leaflet made it simple to explain to parents.

Overwhelmingly, respondents rated SBY as being helpful (77%; 72/93), with only 22% (20/93) reporting they already knew some of the information provided. Topics rated 'very helpful' by most respondents involved: how sleep works (body clock and sleep pressure; 85% (79/93)); how parents' emotional state can affect babies sleep (75% (70/93)); and variability in how long babies need to sleep (71% (66/93)). Practitioners felt that SBY's framing of infant sleep within a wider socio-cultural and biological context was useful for challenging unrealistic parental expectations and for relieving parental stress and anxiety.

> "This information is useful and practical and does challenge some 'traditional' thoughts parents are exposed to" [ID 491499–49001790]

> "The graph that explains sleep pressures is very useful and I have drawn this to explain to parents when leaflet has not been available." [ID 48992634]

> Brilliant practical strategies that will build on the above basics, simple ideas that can be repeated to parents many times without feeling harsh or bossy or unrealistic [ID 49175289]

> "It's a flexible model unlike others which you almost feel cannot be adapted" [ID 49176315]

When considering the SBY information on supporting parents, two topics were rated very helpful by most respondents: how parents can support healthy sleep regulation (setting a wake time, letting in daylight, keeping active during the day; 80% (74/93)); and how parents can respond to their baby's cues to help keep them 'dialed down' (73% (68/93)). Around two thirds of practitioners rated creating positive goals in the parenting journey (66% (61/93)), and mindfulness and preventing unhelpful patterns of thinking (63% (59/93)) as very helpful.

Practitioners felt that the SBY information was realistic, useful and simple for both themselves and parents to understand and implement. SBY offered new information around infant sleep as well as reinforcing existing knowledge, and respondents felt more confident about supporting parents with infant sleep using the SBY discussion tool.

> "I loved [this approach]. It just further supports what we do already. It moves away from sleep training which we don't recommend as a Trust. The terminology used was very good. It helps to change parents' expectations of sleep." [ID 48734830]

> "I feel much more confident at helping parents who are experiencing difficulties with infant sleep." [ID 48721737]

### Practitioner feedback on delivering the intervention to parents

At the time of completing the survey, 81% (75/93) of the respondents had delivered some or all of the SBY information to parents experiencing infant sleep issues: (65% (49/75) provided the information to 1–5 parents, 19% (14/75) to 6–10, and 16% (12/75) to 11 or more) and 90% (84/93) reported feeling confident about sharing the SBY approach with parents. Most practitioners (85% (91/93)) agreed that SBY was straightforward to explain to parents, and that parents understood the information (88% (82/93)). Two respondents commented that for parents whose first language is not English a 'short and to the point' key facts summary would be helpful.

> "Used for the first time recently, whilst discussing the leaflet mum commented on how easy it was to follow and is happy to give it a try." [ID 50200977]

Generally, respondents felt that parents were receptive to SBY (88% (82/93). One commented that "First time parent[s] particularly loved it" [ID 48617672], while another offered a realist assessment that "Sadly some parents do just want a quick fix and even after going through this may resort to leaving a baby crying." [ID 48734830]

Most respondents felt they only had enough time to discuss some of SBY information with parents (79% (73/93)) compared to *all* of the information (29% (27/93)). In free text comments they indicated that, generally, 'core' contacts with parents did not allow sufficient time to comprehensively deliver SBY and some suggested that dedicated SBY sessions would produce the best outcomes.

In both phases of the evaluation practitioners and parents were asked to notify the research team if any actual or anticipated harm was identified or reported arising from information delivered via Sleep, Baby & You; no actual or potential harms were reported.

## Discussion

In developing the intervention materials for SBY we gained further confirmation of a generally unmet need for support around parental fatigue (PPF) and infant-related sleep disruption (IRSD) in the UK. Practitioners working with families in community settings (typically Health Visitors, Community Support Workers, Infant Feeding Support Workers) reported that they were frequently asked by parents for help with sleep issues. In most cases practitioners felt the problem being presented involved parental expectations regarding normal infant sleep, rather than an infant sleep problem requiring treatment. SBY was considered a good fit with responsive based approaches emphasised in current family support, and its flexible and 'common sense' approach was appreciated.

Practitioners reported they lacked the training or tools to effectively help families seeking support around infant-related sleep disruption (IRSD) issues. Most practitioners were familiar with a limited number of behavioural treatment interventions that are variants of the extinction-based approach such as gradual extinction or the disappearing chair, but felt these were inappropriate for young babies, were uncomfortable recommending them, and did not have sufficient training to use them. Practitioners found the SBY approach to be empowering, increasing their confidence in discussing infant sleep with parents.

Feedback from parents confirmed that information and guidance on normal infant sleep was not typically shared by UK health practitioners either antenatally or postnatally, and that this information would be beneficial. Parents reported feeling anxious and confused about infant sleep development and frustrated by culture pressure to sleep train their babies from an early age. Both parents and practitioners felt SBY re-oriented parental expectations around infant sleep from 'when will baby sleep through the night' to 'what should I expect and how can I adapt to it'.

Specific suggestions from field-testing that will be incorporated into future development and testing of this intervention include:

a. The value of offering SBY as a universal anticipatory intervention with antenatal and post-natal delivery; to this end we have developed a discussion tool for use in antenatal classes that aims to help calibrate realistic expectations among parents and encourages them to think ahead to anticipate common scenarios. The level of sleep disruption experienced with a young infant is reportedly surprising to new parents [36] suggesting they may be ill-prepared for its consequences [37], and some anticipatory guidance should be helpful.

b. The option of delivering SBY as a group-based intervention rather than one-to-one in the post-natal period; practitioners felt parents may appreciate the opportunity to discuss SBY in a group setting rather than one to one with a practitioner which was time-consuming. Practitioners also noted that some parents needed the behaviour change suggestions to be drip-fed rather than consumed in a single setting. The opportunity for drop-in groups for support around sleep has been incorporated into the intervention protocol.

c. The development of discussion and information tools that support different learning needs and can be used for self-learning in order to appeal to as wide an audience as possible. To facilitate this we have developed a series of SBY animated videos to explain each page of the discussion tool with voice-over narration. We will also develop an SBY website containing

links to various techniques suggested in SBY such as progressive muscle relaxation and mindfulness.

d.  The ability of families to sustain relevant behaviour changes for a 2-week period does not confirm sustained behaviour change and therefore in future testing of this intervention we propose to record maintenance of behaviour change at 1 month and 4 months post-intervention.

The above modifications and enhancements to SBY made as a consequence of practitioner and parent feedback following field-testing have increased the flexibility and scope of the intervention with a view to increasing the likelihood that it will be suitable for adoption by NHS Trusts in the future.

Although post-partum fatigue and sleep-disruption are common among new parents there are few interventions that aim to support parents and enhance resilience while responsively meeting infant needs across the first year. To date strategies for resolving the effects of normal infant sleep behaviour on parental wellbeing have focussed on 'fixing the baby' via the development and testing of interventions to prolong infant sleep, prevent infants from signalling at night, or treat presumed organic disorders such as Gastro-oesophageal Reflux e.g. [22,38,39]. Although several trials have tested the effectiveness of educational interventions (based on sleep hygiene, self-settling, and the use of routines) alone or in combination with other psychological approaches [13,23–25] they have at best produced mixed results, and most have been found to be ineffective [14,40–42]. 'Limit-setting' educational approaches designed to support the development of 'settled infant night-time behaviour' have been examined as preventative interventions [7,24,26] but have been unsuccessful in achieving their primary outcome of increasing the duration of infant night-time sleep, although researchers do not agree on the strength of evidence for these approaches and better research is needed.

Although it is often stated that behaviourally-based interventions are highly efficacious for resolving infant sleep problems the most robust evidence for their effectiveness comes from trials conducted with children beyond infancy, in controlled settings with direction from a trained professional [27,28]. Behavioural interventions are also time-consuming and complex, and as the majority involve some form of extinction training using operant conditioning they are not only controversial, but developmentally inappropriate for young infants [29], distressing for parents [30], and increase risks of premature breastfeeding cessation, sudden infant death, increased infant crying, and decoupling of mother-infant physiological synchrony [31–33,43]. Although considered a standard component of infant caregiving in many post-industrial countries around the close of the twentieth century, extinction methods have been critiqued because they a) pathologise normal infants and their sleep patterns; b) reinforce unrealistic parental expectations; c) disempower parents and d) do not observe the principles of responsive parenting. These approaches do not assist parents to alter unrealistic expectations, and negative thinking, or empower parents to experiment to discover what works for them and their baby. We agree with Loutzenhiser et al [44] on the need to reconsider the parent-infant sleep relationship, focussing on the family, rather than the infant, and plan to test the effectiveness of SBY via a randomised trial in the near future.

## Conclusions

Helping parents to better understand, and find ways of coping with, their infants' unique sleep needs has the potential to improve parental mental well-being and infant safety while avoiding

inappropriate medicalisation and treatment of infants. 'Sleep, Baby & You' is a promising tool for promoting parental attitude and behaviour-change, that aims to adjust parental expectations and reduce negative thinking around infant sleep, promote responsive infant care in the face of infant-related sleep disruption and fatigue, and support parental well-being during the first year of parenthood. Initial field-testing provided insights useful for further development and subsequent testing via a randomised trial. Support exists for incorporating 'Sleep Baby & You' into an anticipatory, universal intervention to support parents who may experience postpartum fatigue and infant sleep disruption.

## Acknowledgments

We are grateful for the assistance of all parents and practitioners who assisted with the initial scoping meetings, and with this evaluation, and of the service managers who gave permission for field-testing.

We are especially grateful to the members of the SBY working group who volunteered their time and enthusiasm for this project. Steering Committee members were:

Sarah Brooker, Health Visitor and Infant Feeding Co-ordinator, Newcastle upon Tyne Hospitals Trust (email: sarah.brooker@nhs.net); Phyll Buchanan, Peer Counsellor Trainer, Breastfeeding Network; Angela Laverick, Parent & Baby Massage Therapist, My Kid Rocks; Lynne McDonald, Midwife, Newcastle upon Tyne Hospitals Trust; Karen Straughan, Health Visitor, Newcastle upon Tyne Hospitals Trust; Debbie Wade, Health Visiting Lead, Northumberland NHS Trust; and 4 anonymous members.

## Author Contributions

**Conceptualization:** Helen L. Ball, Pamela S. Douglas.

**Data curation:** Catherine E. Taylor.

**Formal analysis:** Helen L. Ball, Catherine E. Taylor.

**Funding acquisition:** Helen L. Ball.

**Investigation:** Helen L. Ball, Catherine E. Taylor, Victoria Thomas, Pamela S. Douglas.

**Methodology:** Helen L. Ball, Catherine E. Taylor, Victoria Thomas.

**Project administration:** Catherine E. Taylor.

**Supervision:** Helen L. Ball.

**Validation:** Helen L. Ball.

**Writing – original draft:** Helen L. Ball, Catherine E. Taylor.

**Writing – review & editing:** Helen L. Ball, Catherine E. Taylor, Victoria Thomas, Pamela S. Douglas.

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
