## [Decision Letter · Decision Letter 0]

22 May 2020

PONE-D-20-03162

Development and evaluation of *Sleep, Baby & You* -- an approach to supporting parental well-being and responsive infant caregiving.

PLOS ONE

Dear Dr. Ball,

Thank you for submitting your manuscript to PLOS ONE. After careful consideration, we feel that it has merit but does not fully meet PLOS ONE’s publication criteria as it currently stands. Therefore, we invite you to submit a revised version of the manuscript that addresses the points raised during the review process.

We look forward to receiving your revised manuscript.

Kind regards,

Elisabete Alves

Academic Editor

PLOS ONE

2.  Please specify whether the T Possums Sleep Program was tested in a randomised controlled trial in Australia.

"HB, CE, VT have no competing interests

PD is Medical Director of Possums Education, a health promotion charity which sells education programs that upskill health professionals in delivery of the Possums Sleep Program. All revenue raised by Possums Education is invested in the development of research and educational materials."

5. One of the noted authors is a group or consortium (SBY working group). In addition to naming the author group, please list the individual authors and affiliations within this group in the acknowledgments section of your manuscript. Please also indicate clearly a lead author for this group along with a contact email address.

6. Your ethics statement must appear in the Methods section of your manuscript. If your ethics statement is written in any section besides the Methods, please move it to the Methods section and delete it from any other section. Please also ensure that your ethics statement is included in your manuscript, as the ethics section of your online submission will not be published alongside your manuscript.

Reviewers' comments:

Reviewer's Responses to Questions

**Comments to the Author**

1. Is the manuscript technically sound, and do the data support the conclusions?

Reviewer #1: Partly

Reviewer #2: No

2. Has the statistical analysis been performed appropriately and rigorously? 

Reviewer #1: Yes

Reviewer #2: No

3. Have the authors made all data underlying the findings in their manuscript fully available?

Reviewer #1: Yes

Reviewer #2: No

4. Is the manuscript presented in an intelligible fashion and written in standard English?

Reviewer #1: Yes

Reviewer #2: Yes

5. Review Comments to the Author

Reviewer #1: Please see attached comments.

Reviewer #2: Dr Douglas, the fourth author in this article, has recently circulated a petition to promote her Possums Sleep Programme approach to infant care in Australia. Supported by the views of parents and practitioners, this article proposes to import the Possums approach into the UK. While there is no doubting Dr Douglas’ commitment to this cause, a distinction has to be drawn between a mission and science. The goal of allowing parents to have choice is a worthy one, but there is a need to meet generic scientific and PLOS ONE criteria for balance, objectivity and safety. Unfortunately, the article as currently written does not meet these criteria. My aim below is to highlight the infant safety and other concerns involved. If the authors consider it is timely and wise to revise the manuscript, the individual points are designed to identify the concerns most in need of attention.

Abstract. ‘Previous research indicates that UK parents would value an approach that facilitates meeting their infants’ needs while supporting their own sleep-related well-being throughout their infant’s first year.’ As this wording makes clear, meeting infants’ needs is a key concern. The rest of this article must make clear where they are met.

Abstract. ‘The intervention … helps parents support the development of their babies’ biological sleep regulators and promote their own well-being.’ This report does not provide any evidence that the Possums Sleep Programme helps to support the development of babies’ biological sleep regulators. The wording is misleading by implying the existence of this evidence, which is an exaggeration. There is some evidence that some parents value the Possums approach and the wording needs to be confined to this and other areas supported by evidence. The ‘Conclusions’ section of the Abstract is more accurate and balanced.

Competing Interests. Dr Douglas does declare an interest. However, I understand that she is financially supported by the Possums programme and may receive a salary from it. If so, it is normal practice in scientific journals to declare these interests.

P5. ‘Treatment-oriented interventions often involve extinction techniques applied to infants who are causing parental sleep disruption (22,23,25–27). Examples of these approaches are summarised in Table 1’. These examples are biased, not representative. Two of them appear to have been chosen because of negative findings. It is true that such findings exist, but the majority of studies and systematic reviews show these methods to be effective in reducing infant night waking and crying out and enhancing infant sleep. The authors do themselves no favours by ignoring this evidence: see, e.g. Kempler et al. Sleep Medicine Reviews 29 (2016) 119e120. Also, Hiscock & Fisher Journal of Paediatrics and Child Health 51 (2015) 361–364. Table 1 needs to contain representative examples.

P7. ‘The materials will be used as the basis for an intervention intended to reduce the proportion of parents experiencing postpartum depression, PPF and ISD and support to parents on meeting the needs of their infants…’ ISD is defined above as Infant-related Sleep Disruption. It needs to be made explicit that this involves parental sleep disruption and called IRSD, or PSD, since in the literature ISD is more commonly an abbreviation for Infant Sleep Disturbances (or Disorders). The authors need to make it unambiguous throughout their article that they are not claiming to change infant sleep-waking behaviors – and that there is no evidence that Possums does so or meets the needs of infants.

P8. The Possums programme is also called ‘Neuroprotective Developmental Care (NDC) by its providers, but to the best of my knowledge there is no evidence that it protects neurological integrity or infant development. For this scientific report, there is no need for the phrase Neuroprotective Developmental Care and it would be better to omit it.

P8. ‘The program educates parents in basic sleep science and how this relates to their infant’s sleep development. It offers strategies for optimising health function of the biological sleep regulators, to protect against excessive night-waking. It supports values-clarification and empowers parents to adjust their expectations and to experiment with how they might meet their babies’ needs. It supports cued care according to the most recent neuroscience and psychological attachment research, with the aim of promoting secure attachment, adequate sleep, and parent-infant enjoyment (34)‘ The reference given is to a small scale pilot study carried out by some authors of this article. That study did not provide any evidence that the Possums programme meets infant needs, or that it promotes adequate infant sleep or secure attachment. Here too the authors need to be unambiguous and evidence-based in their writing.

P 8-9. ‘The parent evaluation and Ball’s observations of the PSP in practice confirmed that the approach taken by this programme was consistent with a responsive approach and would be appropriate for development into a UK-based intervention.’. I think this is just saying that the first author approved it. In what way did scientific evidence confirm that the programme ‘would be appropriate for development into a UK-based intervention’? This sentence would be better omitted.

P9. ‘Stakeholders confirmed there was a gap in UK service provision addressing parental fatigue and sleep disruption during infancy.’ Yes, so far as this report is clear that Possums focuses on parental needs it is on firmer ground.

P9. ‘As most community services in the region encouraged responsive parenting, extinction methods were considered inappropriate.’ Who considered them inappropriate? We need firm scientific evidence here. Most current health system and clinical sources I’m aware of recommend limited versions of what Hiscock and Fisher (see above) call infant behavioural management. Unless it can be supported, this claim should be omitted.

Table 2. What does ‘dial baby down’ mean?

P14. ‘Practitioners felt SBY also empowered and equipped … and they appreciated having a non-sleep training option to offer parents.’ Yes, supporting evidence-informed choice is desirable providing infant and parental health and safety are safeguarded. However, parents also need to be informed about the evidence for sleep training in an unbiased way. They can then choose according to their goals and priorities. It isn’t necessary to promote the Possums approach as the universal solution for all infants and parents.

P 14. ‘I’m using it all the time, I think it’s amazing, there’s nothing that you teach that could do any harm so why wouldn’t you implement it?’ I appreciate that this is a participant comment, but it does get to the heart of the matter. Please see the final point below.

P16. This needs to specify how many individual parents provided data and how many were mums or dads. The information above about families leaves this unclear.

P17. ‘Improvements in frequent night waking and feeling stressed about their baby’s sleep were the most frequently reported positive outcomes.’ How many parents reported this? Please make clear: was the improvement in the parents’ night waking, their infant’s night waking behaviour, or the extent to which their infant’s behaviour was a problem for parents?

P19. ‘Practitioners appreciated that SBY did not employ sleep training methods,’ how many – numbers and percentages are needed here.

P21. Discussion. ‘infant sleep disruption’ Having read this article through, I think this phrase is unhelpful and needs to be replaced. It is unclear whether the disruption involves infant or parental sleep and this ambiguity leads to a lack of clarity in the authors’ writing and potentially in readers’ understanding. ‘Infant-related Parent Sleep Disruption’ or simply Parent Sleep Disruption would be unambiguous. The phrase ‘Infant Sleep Disruption’ should be confined to measures of infants.

P23. ‘Although several trials have tested the effectiveness of educational interventions (based on sleep hygiene, self-settling, and the use of routines) alone or in combination with other psychological approaches (13,23–25) they have at best produced mixed results, and most have been found to be ineffective 1 (14,40–42) ‘Limit-setting’ educational approaches designed to support the development of ‘settled infant night-time behaviour’ have been examined as preventative interventions (7,24,26) but have been unsuccessful in achieving their primary outcome of increasing the duration of infant night time sleep.’ It is true that findings are mixed, but I believe this claim that most studies have found these methods to be ineffective is inaccurate. As noted above, I am not alone in that conclusion. More and better research is needed, but disregarding the published systematic reviews and meta-analyses is not scientific or helpful.

Overall, it is hard to disagree with the general point that many parents have unrealistic expectations for their baby’s sleep. The solution for that, though, is to provide accurate information. Although the evidence for the Possums approach is very limited, it appears that its primary benefits are for parents, particularly mums and particularly mums who can adapt to their baby’s sleep-waking habits. That is challenging to manage where both parents work office hours. There is firm evidence that infant crying which overwhelms parents can trigger shaken baby syndrome and other forms of infant abuse, raising safety concerns.

Many parents and practitioners will have seen how distressed and wretched sleep-deprived babies become. Until Possums collects and provides measurements of infant sleep-waking and distress, the concern will remain that it helps infants less than the ‘infant sleep management methods’ referred to above. There is robust evidence from two genetic studies that the home environment – parenting not genes - makes the main contribution to individual differences in infant sleep-waking behaviour and problems (see e.g. Fisher et al.: www.pediatrics.org/cgi/doi/10.1542/peds.2011-1571). We know from the Gemini twin study that some infants need parental support to help them develop healthy eating habits. It would be a big surprise if that is not also true for infant sleep-waking behaviour.

A basic principle in both science and medicine is to do no harm. Archie Cochrane of Cochrane reviews fame developed that principle into the axiom that a new treatment should be introduced only when it is clear that it will do good and no more harm than existing treatments. My own view is that this article is premature. The priority is for the Possums approach to measure and publish evidence of any harms and benefits this approach has for infant sleep-waking and development more generally. There are simple parent-report methods for measuring infant sleep-waking and distress which, while not perfect, would provide evidence suitable for this purpose. It is puzzling that these authors have not used them. In the meantime, any findings published need to keep Cochrane’s principle in mind and to conform to scientific requirements for balance, objectivity and safety.

6. PLOS authors have the option to publish the peer review history of their article (what does this mean?). If published, this will include your full peer review and any attached files.

Reviewer #1: No

Reviewer #2: No

---

## [Author Response · Author response to Decision Letter 0]

1 Jul 2020

Please see Response to Reviewers document uploaded

---

## [Editor Report · Decision Letter 1]

23 Jul 2020

Development and evaluation of 'Sleep, Baby You' -- an approach to supporting parental well-being and responsive infant caregiving.

PONE-D-20-03162R1

Dear Dr. Ball,

We’re pleased to inform you that your manuscript has been judged scientifically suitable for publication and will be formally accepted for publication once it meets all outstanding technical requirements.

Kind regards,

Elisabete Alves

Academic Editor

PLOS ONE